# Influence of Socioeconomic Status on the Presence of Obstructive Coronary Artery Disease and Cardiovascular Outcomes in Patients Undergoing Invasive Coronary Angiography

**DOI:** 10.3390/healthcare12020228

**Published:** 2024-01-17

**Authors:** Jaehoon Chung, Woo-Hyun Lim, Hack-Lyoung Kim, Hyun Sung Joh, Jae-Bin Seo, Sang-Hyun Kim, Joo-Hee Zo, Myung-A Kim

**Affiliations:** 1Division of Cardiology, Department of Internal Medicine, National Medical Center, Seoul 04564, Republic of Korea; drjhchung1@gmail.com; 2Division of Cardiology, Department of Internal Medicine, Boramae Medical Center, Seoul National University College of Medicine, Seoul 07061, Republic of Korea; woosion@gmail.com (W.-H.L.); wingx4@naver.com (H.S.J.); cetuximab@naver.com (J.-B.S.); shkimmd@snu.ac.kr (S.-H.K.); jooheezo@hanmail.net (J.-H.Z.); kma@snu.ac.kr (M.-A.K.)

**Keywords:** cardiovascular disease, coronary angiography, coronary artery disease, medical aid beneficiary, national health insurance beneficiary, socioeconomic status

## Abstract

This retrospective study investigated the impact of socioeconomic status (SES) on patients at high risk of cardiovascular disease, focusing on obstructive coronary artery disease (CAD) presence and long-term cardiovascular outcomes in individuals undergoing invasive coronary angiography (ICA). Analyzing data from 9530 patients categorized by health insurance type (medical aid beneficiaries (MABs) as the low SES group; national health insurance beneficiaries (NHIBs) as the high SES group), this research explores the relationship between SES and outcomes. Despite a higher prevalence of cardiovascular risk factors, the MAB group exhibited similar rates of obstructive CAD compared to the NHIB group. However, over a median 3.5-year follow-up, the MAB group experienced a higher incidence of composite cardiovascular events, including cardiac death, acute myocardial infarction, coronary revascularization, and ischemic stroke, compared with the NHIB group (20.2% vs. 16.2%, *p <* 0.001). Multivariable Cox regression analysis, adjusting for potential confounders, revealed independently worse clinical outcomes for the MAB group (adjusted odds ratio 1.28; 95% confidence interval 1.07–1.54; *p* = 0.006). Despite comparable CAD rates, this study underscores the fact that individuals with low SES encounter an elevated risk of composite cardiovascular events, emphasizing the association between socioeconomic disadvantage and heightened susceptibility to cardiovascular disease, even among those already at high risk.

## 1. Introduction

Cardiovascular disease (CVD) remains the leading cause of death worldwide and is a huge contributor to economic burden despite the advancement of many excellent diagnoses and treatments. The prevalence of CVD continues to be high due to the increasing life span in aging societies, and the socioeconomic costs are also continuously increasing [1,2]. Many previous studies have identified traditional risk factors for CVD such as hypertension, dyslipidemia, diabetes, a family history of premature coronary artery disease (CAD), and smoking [3]. Recent studies have shown that socioeconomic status (SES) is also a factor related to CVD [4,5,6]. Although numerous studies on the relationship between SES and CVD have been conducted in the general population [4,5,6], studies on the association between SES and cardiovascular outcomes in high-risk patients are scarce. Cardiovascular events are more likely to occur in high-risk patients, so it is very important to identify their risk factors. Therefore, the purpose of this study was to investigate whether there is a difference in the presence and extent of coronary artery stenosis in invasive coronary angiography (ICA) according to SES in patients at high CVD risk. Since CAD severity is closely related to the occurrence of future cardiovascular events, we also aimed to compare the prognosis according to SES.

## 2. Materials and Methods

### 2.1. Study Subjects

Between May 2008 and May 2020, consecutive patients aged 20 to 90 years with suspected CAD who underwent ICA were retrospectively analyzed. There was no clinical limitation to enrollment except for patients diagnosed with cancer within the last five years. Finally, a total of 9530 patients were analyzed in this study. This study was conducted in accordance with the Declaration of Helsinki, and the study protocol was approved by the Institutional Review Board (IRB) of Boramae Medical Center (Seoul, Republic of Korea) (IRB number 10-2021-43, approval on 1 April 2021). Obtaining written informed consent was waived by the IRB due to the routine nature of the information collected and the retrospective nature of the study design.

### 2.2. Data Collection

Each patient’s height and body weight were measured at the time of admission. Body mass index (BMI) was calculated as weight in kilograms divided by the square of height in meters (kg/m^2^). A history of cardiovascular risk factors, including hypertension, diabetes mellitus, dyslipidemia, and current cigarette smoking status, was obtained. Previous clinical medical history, including myocardial infarction, coronary revascularization, heart failure, and stroke, was also assessed. Venous blood samples were obtained after overnight fasting for more than 12 h, and the following laboratory parameters were measured: hemoglobin, glycated hemoglobin (HbA1c), C-reactive protein, creatinine, total cholesterol, low-density lipoprotein (LDL) cholesterol, high-density lipoprotein (HDL) cholesterol, and triglyceride. The estimated glomerular filtration rate (eGFR) was calculated by using the Modification of Diet in Renal Disease (MDRD) study equation. Transthoracic echocardiography was performed during hospitalization, and left ventricular ejection fraction (LVEF) was measured with the biplane method of disks (modified Simpson’s rule) according to current practice guidelines [7]. Information on current cardiovascular medications, including aspirin, clopidogrel, statins, angiotensin-converting enzyme inhibitors, angiotensin receptor blockers, beta-blockers, and calcium channel blockers, was also obtained.

### 2.3. ICA and Percutaneous Coronary Intervention

When performing ICA, the access site was determined at the operator’s discretion, regardless of whether it was transradial or transfemoral ICA. Obstructive CAD was defined as ≥50% stenosis in major coronary arteries or their branches with >2 mm diameter on ICA, and the extent of CAD is based on the number of affected coronary arteries. Percutaneous coronary intervention (PCI) was performed according to current procedural guidelines [8,9,10,11].

### 2.4. SES

The healthcare system in Korea is largely divided into national health insurance (NHI) and medical aid (MA), with private medical insurance serving a supplementary function. NHI is a public health insurance program run by the Ministry of Health and Welfare that requires citizens with sufficient income to pay contributions to insure themselves and their dependents. The MA system is a public assistance system that the state guarantees for medical problems (disease, injury, childbirth, etc.) of low-income people who do not have the ability to sustain life or have difficulties in living. Medical aid beneficiaries (MABs) are households with less than 40% of the standard median income, households unable to work, homeless people, facility recipients, victims of loss, and defectors from North Korea [12,13]. Eligible amounts of household income for MAB are shown in Appendix A. In previous studies, compared to NHIB, MAB had lower income, lower education level, less participation in public pension, and lower private health insurance coverage [14,15]. In this study, the patients were divided into two groups according to the health insurance type: those with low SES who had the MA program (MAB group; n = 1436) and those with high SES who had the NHI program (NHIB group; n = 8094).

### 2.5. Clinical Events

The primary outcome was a major adverse cardiac and cerebral event (MACCE), which is a composite of cardiac death, non-fatal myocardial infarction, coronary revascularization, and non-fatal ischemic stroke during clinical follow-up. All deaths were considered to be related to cardiac causes unless a clear non-cardiac cause was identified. Myocardial infarction was defined as a rise in cardiac troponin with at least one value above the 99th percentile upper reference limit and with at least one of the following: (1) symptoms of ischemia, (2) new or presumed new significant ST-segment–T wave changes or new left bundle branch block, (3) development of pathological Q waves in the electrocardiogram, or (4) imaging evidence of new loss of viable myocardium or new regional wall motion abnormality [16,17]. Coronary revascularization was defined as revascularization with PCI or coronary artery bypass grafting. Ischemic stroke was defined as an episode of neurological dysfunction caused by focal cerebral infarction based on objective evidence of cerebral ischemic injury in a defined vascular distribution [18].

### 2.6. Statistical Analysis

The results are expressed as the mean ± standard deviation (SD), median (IQR) for continuous variables, and as a percentage for categorical variables. Between the two groups, continuous variables were compared using Student’s *t*-test or Wilcoxon rank sum test according to the normality assumption, and categorical variables were compared using Pearson’s chi-square test. The study endpoints were demonstrated using the Kaplan–Meier survival curve and compared using the log-rank test. Multivariable Cox proportional hazards models were used to determine whether medical insurance status was independently associated with clinical outcomes. Age, sex, BMI, hypertension, diabetes mellitus, smoking status, presence of obstructive CAD, LVEF, and the use of concomitant medications, including antiplatelets, renin–angiotensin system blockers, beta-blockers, and statins, were included as covariates in the multivariable models. To determine whether the effect of medical insurance status differs depending on the presence or absence of CAD, clinical outcomes were also analyzed depending on the presence or absence of CAD. Global chi-square values were calculated to clarify the incremental prognostic value of insurance type in combination with other risk factors for predicting future MACCE. All analyses were 2-tailed, and clinical significance was defined as *p <* 0.05. Statistical analyses were performed with the statistical package SPSS version 23.0 (IBM Co., Armonk, NY, USA).

## 3. Results

### 3.1. Baseline Characteristics and ICA-Associated Findings

The mean age of study subjects was 66.0 ± 12.3 years, and 5731 (60.2%) were male. The baseline clinical characteristics of the study subjects according to the health insurance type are shown in Table 1. Of the study patients, 1436 (15.1%) were MABs. The MAB group was older (67.4 ± 11.7 vs. 65.7 ± 12.3 years; *p <* 0.001) and a smaller proportion were male (57.2% vs. 60.8%; *p* = 0.012) compared to the NHIB group. The MAB group had more cardiovascular risk factors, including hypertension, diabetes, and cigarette smoking, compared to the NHIB group (*p <* 0.05 for each). Heart failure history was also more frequent in the MAB group than in the NIHB group. The diagnosis of acute myocardial infarction was more frequent in the NIHB group than in the MAB group. For major laboratory findings, hemoglobin and eGFR were lower, and glycated hemoglobin and C-reactive protein were higher in the MAB group than in the NHIB group. Mean LVEF was lower in the MAB group compared to the NHIB group. Among cardiovascular medications, statins were more frequently prescribed in the NHIB group, and angiotensin-converting enzyme inhibitors and beta-blockers were more frequently prescribed in the MAB group.

The ICA findings according to insurance status are shown in Table 2. Of the total study subjects, 6100 (64.0%) had obstructive CAD. Overall, the prevalence and extent of obstructive CAD were not different between the two groups. The prevalence of left main disease and three-vessel disease was also similar between the two groups. The MAB group received PCI less frequently than the NHIB group (40.0% vs. 43.9%; *p* = 0.006). The PCI results were similar between the two groups.

### 3.2. Clinical Events

The incidence of clinical events during the follow-up is shown in Table 3. During the median follow-up period of 3.5 years (interquartile range, 1.0–5.9 years), there were 1598 (16.8%) MACCEs. The incidence rate of MACCE was significantly higher in the MAB group than in the NHIB group (20.2% vs. 16.2%, *p <* 0.001), which was also confirmed in the Kaplan–Meier survival curves (log-rank *p <* 0.001) (Figure 1). In each component of MACCE, the incidence rates of cardiac death, myocardial infarction, and ischemic stroke were not different between the two groups, but the rate of revascularization was significantly higher in the MAB group than in the NHIB group (10.8% vs. 6.9%, *p <* 0.001). In the multivariable Cox regression analysis, medical insurance status was independently associated with the occurrence of MACCE during the follow-up period even after adjustment for multiple confounding factors (MAB vs. NHIB groups: hazard ratio (HR), 1.28; 95% confidence interval (CI), 1.07–1.54; *p* = 0.006) (Table 4). Additionally, insurance status, age, diabetes mellitus, current smoking, reduced LVEF, and the presence of obstructive CAD were associated with the occurrence of MACCE. To determine whether the effect of medical insurance status differs depending on the presence or absence of CAD, clinical events were also analyzed depending on the presence or absence of CAD. The incidence rate of MACCE was significantly higher in the MAB group than in the NHIB group, regardless of the presence or absence of CAD at baseline (log-rank *p* = 0.017 in patients without CAD, log-rank *p <* 0.001 in those with CAD) (Figure 2). In subgroups with (MAB vs. NHIB groups: HR, 1.24; 95% CI, 1.02–1.52; *p* = 0.035) and without (MAB vs. NHIB groups: HR, 1.52; 95% CI, 1.01–2.29; *p* = 0.047) CAD at baseline, medical insurance status was independently associated with the occurrence of MACCE during the follow-up period (Table 5). The global chi-square values of Cox regression analyses were compared to evaluate the incremental prognostic value of insurance status when it was added to age, sex, and traditional clinical factors, including diabetes mellitus, hypertension, dyslipidemia, and smoking status. The addition of traditional clinical risk factors to age and sex had incremental prognostic value for predicting MACCE (global chi-square value, from 136.6 to 278.4; *p <* 0.001). The addition of SES to combined use of age, sex, and traditional clinical factor results further increased the predictive power for MACCE (global chi-square value, from 278.4 to 284.9; *p* < 0.001) (Figure 3).

## 4. Discussion

The main finding of this study is that MABs had worse clinical outcomes than NHIBs in patients who underwent ICA. Although MABs were older and had more cardiovascular risk factors than NHIBs, the prevalence and extent of CAD were similar between the two groups. This study provided additional evidence for a relationship between low SES and increased CVD risk in a high-risk population.

Many previous studies have reported that low SES is associated with an increased risk of developing CVD and death. In a previous study of the Atherosclerosis Risk in Communities (ARIC) cohort consisting of the general population aged 45 to 64 years in four US communities and Finnish population-based cardiovascular risk factor-monitoring cohorts (FINRISK), low income was significantly associated with increased risk of cardiac events and sudden cardiac death [19]. In another study using ARIC data from more than 15,000 adults, low SES (low education and/or low income) was associated with older age, higher blood pressure, higher total cholesterol, more cigarette smoking, and more frequent use of antihypertensive medications. Thus, subjects with low SES had a higher Framingham risk score than those with high SES. In that study, low SES was associated with an increased coronary heart disease risk, independent of baseline Framingham risk score and time-dependent variables [20]. In an interesting computer simulation study of US adults (35 to 64 years) using data from the 2011–2016 US National Health and Nutrition Examination Survey, both men and women in the low-SES group had double the rates of myocardial infarctions and cardiac deaths per 10,000 person-years compared to those in the high-SES group. The higher burden of traditional CAD risk factors in adults with low SES accounted for only 40% of these excess events. The remaining 60% of these events were attributed to other factors related to low SES [21]. In contrast to previous studies conducted in the general population, we studied the role of SES in high-risk patients with suspected CAD. In our study, similar to previous studies with the general population, low SES was associated with an increased risk of CVD in high-risk patients who underwent ICA for suspected CAD, even after adjusting for baseline risk factors. Also, low SES was associated with an increased cardiovascular risk in both patients with and patients without obstructive CAD. The results of this study suggest that SES may be one of the significant risk factors for CVD, irrespective of baseline cardiovascular risk.

The exact mechanism by which SES acts as a risk factor for CVD is unclear, but there have been some studies that were able to infer the mechanism. According to a study with the Korean National Health and Nutrition Examination Survey, low SES was associated with an increased risk of metabolic syndrome and its components in the Korean adult population [22]. Similarly, MABs had more cardiovascular risk factors, such as hypertension, diabetes, and smoking, in our study. In a Minnesota heart survey study, education level was inversely related to blood pressure, cigarette smoking, and BMI. In addition, a higher education level was associated with an increased time spent in leisure physical activities and health knowledge [23]. Lack of education can lead to risky decisions about health behaviors such as smoking and alcohol consumption. This suggests that SES is associated with modifiable behavioral risk factors. However, modifiable behavioral risk factors alone cannot explain the association between SES and CVD risk. The association between low SES and CVD risk is reinforced by an additional independent effect of SES. Another study suggested that low parental SES or low SES in childhood is associated with the onset of early CVD, independent of traditional risk factors. In a longitudinal cohort study of 1337 medical students at the Johns Hopkins University School of Medicine, lower SES in childhood was associated with a higher risk of early coronary heart disease in adulthood among men with high adulthood SES [24]. As another factor, chronic psychosocial stress associated with low SES promotes atherosclerosis and CAD events. In the Whitehall II study with British civil servants, psychosocial work characteristics and social support were associated with CAD incidence [25]. A systematic review of prospective cohort studies found that social isolation and lack of social support were associated with a two- to three-fold increased risk of cardiac mortality and morbidity [26]. In another study conducted in Denmark with more than 1,660,000 subjects, job strain was associated with a higher risk of developing CHD [27]. In addition to individual SES, socioeconomic characteristics of neighborhoods are associated with CVD risk factors, adverse events, and mortality. In the analysis using data from the Atherosclerosis Risk in Communities study, living in disadvantaged areas is associated with an increased incidence of CHD after controlling for individual income, education, and occupation [28]. Collectively, low SES increases behavioral and psychosocial risks, and these disparities also occur as a result of poor health during childhood and early life, parental risk, and surrounding environments.

The poor prognosis of CVD in MABs is probably explained by the above reasons. This study did not enroll all MABs or NHIBs. However, in patients with suspected CAD who underwent ICA, baseline CAD status could be similar. Moreover, baseline underlying risk factors, such as hypertension, diabetes, and cigarette smoking, were more frequent in the MAB group than in the NHIB group. Although these risk factors were statistically adjusted, there may be a strong possibility that the control status of each patient’s risk factors and the duration of these risk factors would act as unadjusted factors. Besides this, interactions between behavioral and psychosocial risk, disparities during childhood, parental risk, and surrounding environments could lead to worse CVD outcomes.

### 4.1. Clinical Implications

Subjects with low SES are more likely to develop CVD and to have a higher frequency of related risk factors, so CVD tests and treatments should be performed more aggressively. However, approaches only to clinical risk factors paradoxically increase the disparity as groups with higher SES are more likely to receive interventions in a healthcare setting. Therefore, appropriate risk stratification of low-SES patients with traditional risk factors for CVD is important for identifying high-risk patients. Although clear criteria for SES have not yet been established, SES should be considered a CVD risk factor. Furthermore, as modifiable behavioral factors and childhood disparities could influence future CVD outcomes, it is necessary to ensure proper healthcare and health education in childhood and to resolve social inequality caused by low SES.

### 4.2. Study Limitations

This study has several limitations. First, as this study is a single-center study in Korea, it is difficult to generalize our results to other races and populations. Secondly, as this study is of retrospective design, there is a possibility that clinical event information may be inaccurate. Thirdly, even within the same MAB or NHIB groups, education level, income, and social position can vary widely among individuals. Fourthly, our study data did not include specific details for each factor of SES, such as income and education levels, and lifestyle factors. Fifthly, our study revealed that hypertension, a well-known CVD risk factor, was not associated with MACCE occurrence in the multivariable analysis. The potential inaccuracy in defining hypertension due to the retrospective design of this study, along with the influence of antihypertensive medications, might have contributed to these unexpected results. Finally, changes in cardiovascular risk factors and medications were not investigated during the follow-up period.

## 5. Conclusions

Although CAD prevalence was similar, MABs showed an increased risk of composite cardiovascular events compared to NHIBs in Korean adults undergoing ICA. This provides additional evidence for the association between low SES and an increased risk of CVD even in populations at high risk. Cardiologists need an elaborate long-term strategy to reduce cardiovascular risk factors in low-SES patients.

## Figures and Tables

**Figure 1 healthcare-12-00228-f001:**
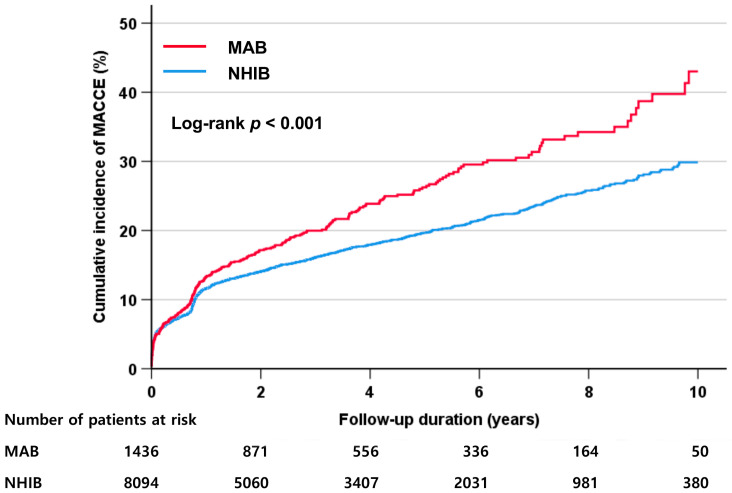
The incidence of MACCE according to the health insurance type. MACCE, major adverse cardiac and cerebral event; MAB, medical aid beneficiary; NHIB, national health insurance beneficiary.

**Figure 2 healthcare-12-00228-f002:**
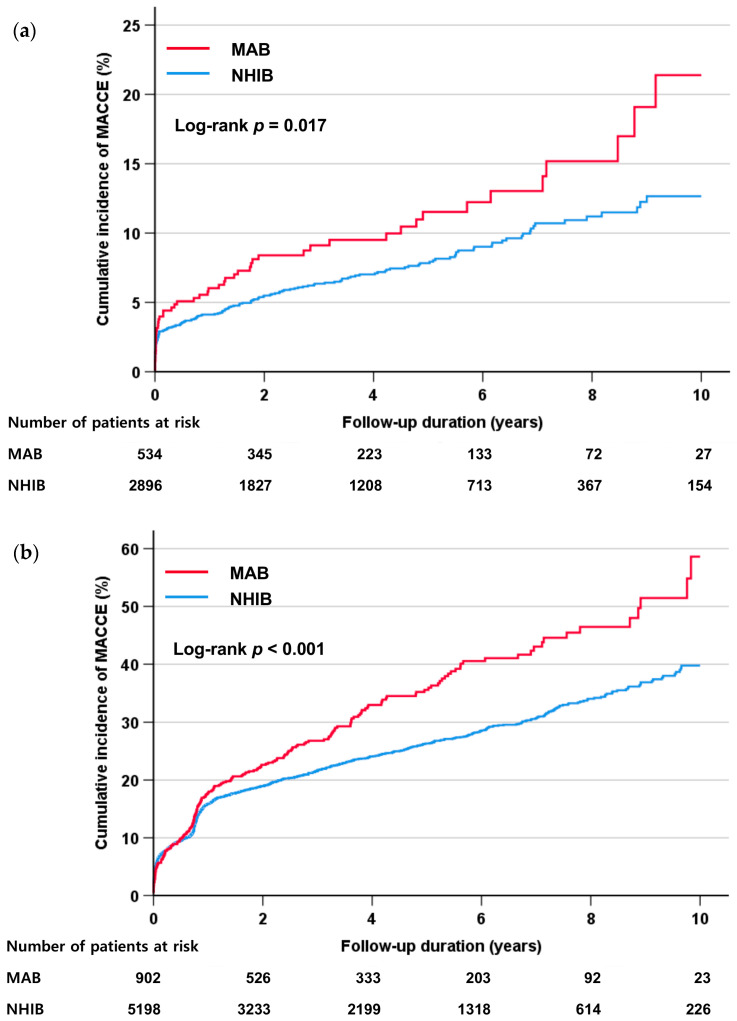
The incidence of MACCE in subjects with or without obstructive coronary artery disease according to the health insurance type. (**a**) Subjects without obstructive coronary artery disease, (**b**) subjects with obstructive coronary artery disease. MACCE, major adverse cardiac and cerebral event; MAB, medical aid beneficiary; NHIB, national health insurance beneficiary.

**Figure 3 healthcare-12-00228-f003:**
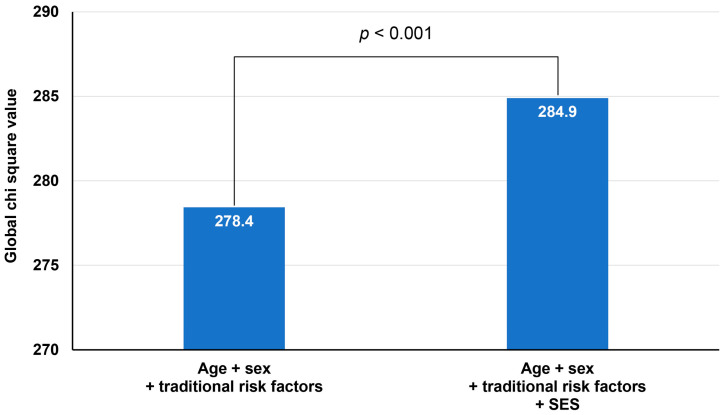
Prognostic value of clinical risk factors and health insurance type for MACCE. MACCE, major adverse cardiac and cerebral event; SES, socioeconomic status.

**Table 1 healthcare-12-00228-t001:** Baseline characteristics of study subjects.

Characteristic	MAB (n = 1436)	NHIB (n = 8094)	*p* Value
Age, years	69 [59–76]	67 [58–75]	<0.001
Male	821 (57.2)	4910 (60.7)	0.012
Body mass index, kg/m^2^	24.0 [21.7–26.6]	24.4 [22.2–26.6]	0.002
Cardiovascular risk factors			
Hypertension	937 (65.3)	4797 (59.3)	<0.001
Diabetes mellitus	564 (39.3)	2540 (31.4)	<0.001
Dyslipidemia	750 (52.2)	4478 (55.3)	0.030
Cigarette smoking	444 (30.9)	1947 (24.1)	<0.001
Obesity (body mass index ≥ 25 kg/m^2^)	535 (37.3)	3234 (40.0)	0.087
Medical history			
Myocardial infarction	81 (5.6)	392 (4.8)	0.200
Coronary revascularization	112 (7.8)	519 (6.4)	0.051
Heart failure	109 (7.6)	448 (5.5)	0.002
Stroke	133 (9.3)	656 (8.1)	0.143
Clinical diagnosis			
Acute myocardial infarction	261 (18.1)	1760 (21.7)	0.002
Major laboratory findings			
Hemoglobin, g/dL	12.6 [11.0–13.8]	13.2 [11.8–14.4]	<0.001
Glycated hemoglobin, %	6.2 [5.7–7.2]	6.1 [5.7–7.0]	0.001
C-reactive protein, mg/dL	0.3 [0.1–1.0]	0.2 [0.1–0.7]	<0.001
Estimated GFR, mL/min/1.73 m^2^	72.0 [52.3–88.6]	78.3 [62.1–93.0]	<0.001
Total cholesterol, mg/dL	154 [128–185]	158 [132–189]	0.004
LDL cholesterol, mg/dL	89.2 [66.1–115.9]	90.2 [68–117.4]	0.454
HDL cholesterol, mg/dL	41 [34–50]	42 [35–50]	0.351
Triglyceride, mg/dL	107 [76–152]	105 [78–146]	0.824
Left ventricular ejection fraction, %	63.0 [51.0–68.2]	64.6 [57.0–68.9]	<0.001
Cardiovascular medications			
Aspirin	1231 (85.7)	6840 (84.5)	0.238
Clopidogrel	1124 (78.3)	6204 (76.6)	0.179
Statins	713 (49.7)	4250 (52.5)	0.046
Angiotensin-converting enzyme inhibitors	163 (11.4)	754 (9.3)	0.016
Angiotensin receptor blockers	404 (28.1)	2114 (26.1)	0.110
Beta-blockers	498 (34.7)	2521 (31.1)	0.008
Calcium channel blockers	492 (34.3)	2546 (31.5)	0.035

Numbers are expressed as mean ± standard deviation, median (interquartile range, IQR), or n (%). MAB, medical aid beneficiary; NHIB, national health insurance beneficiary; GFR, glomerular filtration rate; HDL, high-density lipoprotein; LDL, low-density lipoprotein.

**Table 2 healthcare-12-00228-t002:** ICA-associated findings.

Characteristic	MAB (n = 1436)	NHIB (n = 8094)	*p* Value
Obstructive CAD	902 (62.8)	5198 (64.2)	0.306
CAD extent			0.045
Insignificant	534 (37.2)	2896 (35.8)	
One-vessel disease	253 (17.6)	1616 (20.0)	
Two-vessel disease	251 (17.5)	1530 (18.9)	
Three-vessel disease	398 (27.7)	2052 (25.4)	
Left main disease	94 (6.5)	550 (6.8)	0.729
Three-vessel disease	398 (27.7)	2052 (25.4)	0.059
Percutaneous coronary intervention	574 (40.0)	3553 (43.9)	0.006
Procedure results			0.551
Successful	535 (93.2)	3287 (92.5)	
Suboptimal	13 (2.3)	110 (3.1)	
Failed	26 (4.5)	156 (4.4)	

Numbers are expressed as n (%). ICA, invasive coronary angiography; MAB, medical aid beneficiary; NHIB, national health insurance beneficiary; CAD, coronary artery disease.

**Table 3 healthcare-12-00228-t003:** Clinical events.

Characteristic	MAB (n = 1436)	NHIB (n = 8094)	*p* Value
MACCE	290 (20.2)	1308 (16.2)	<0.001
Cardiac death	87 (6.1)	478 (5.9)	0.821
Myocardial infarction	24 (1.7)	134 (1.7)	0.966
Coronary revascularization	155 (10.8)	560 (6.9)	<0.001
Ischemic stroke	46 (3.2)	203 (2.5)	0.128

Numbers are expressed as n (%). MACCE, major adverse cardiac and cerebrovascular event; MAB, medical aid beneficiary; NHIB, national health insurance beneficiary.

**Table 4 healthcare-12-00228-t004:** Clinical factors associated with composite major cardiac and cerebrovascular events.

Variable	HR	95% CI	*p* Value
MAB (vs. NHIB)	1.28	1.07 to 1.54	0.006
Age ≥ 65 years	1.59	1.35 to 1.86	<0.001
Female sex	0.93	0.80 to 1.08	0.346
BMI ≥ 25 kg/m^2^	0.97	0.84 to 1.11	0.624
Diabetes mellitus	1.49	1.30 to 1.71	<0.001
Hypertension	1.07	0.91 to 1.25	0.400
Current smoking	1.26	1.06 to 1.49	0.008
LVEF < 50%	1.62	1.37 to 1.92	<0.001
Obstructive CAD	3.13	2.58 to 3.79	<0.001
Antiplatelets	0.92	0.72 to 1.19	0.543
Statins	0.90	0.78 to 1.04	0.162
Beta-blockers	1.03	0.90 to 1.18	0.686
RAS blockers	1.09	0.94 to 1.26	0.244

HR, hazard ratio; CI, confidence interval; MAB, medical aid beneficiary; NHIB, national health insurance beneficiary; BMI, body mass index; LVEF, left ventricular ejection fraction; CAD, coronary artery disease; RAS, renin–angiotensin system.

**Table 5 healthcare-12-00228-t005:** Clinical factors associated with composite major cardiac and cerebrovascular events according to presence of obstructive CAD.

Variable	Without CAD (n = 3430)			With CAD (n = 6100)		
	HR	95% CI	*p* Value	HR	95% CI	*p* Value
MAB (vs. NHIB)	1.52	1.01 to 2.29	0.047	1.24	1.02 to 1.52	0.035
Age ≥ 65 years	2.22	1.52 to 3.26	<0.001	1.47	1.23 to 1.75	<0.001
Female sex	1.04	0.71 to 1.52	0.829	0.93	0.79 to 1.10	0.380
BMI ≥ 25 kg/m^2^	0.79	0.55 to 1.14	0.213	1.00	0.86 to 1.17	0.963
Diabetes mellitus	1.55	1.07 to 2.26	0.022	1.50	1.29 to 1.74	<0.001
Hypertension	1.16	0.79 to 1.72	0.449	1.05	0.89 to 1.25	0.550
Current smoking	1.57	1.00 to 2.47	0.050	1.21	1.01 to 1.46	0.037
LVEF < 50%	2.41	1.60 to 3.63	<0.001	1.55	1.29 to 1.86	<0.001
Antiplatelets	0.51	0.35 to 0.74	<0.001	1.45	1.00 to 2.11	0.052
Statins	0.70	0.48 to 1.01	0.053	0.93	0.80 to 1.09	0.398
Beta-blockers	0.78	0.53 to 1.14	0.199	1.06	0.91 to 1.23	0.447
RAS blockers	1.25	0.86 to 1.83	0.243	1.05	0.90 to 1.22	0.568

CAD, coronary artery disease; HR, hazard ratio; CI, confidence interval; MAB, medical aid beneficiary; NHIB, national health insurance beneficiary; BMI, body mass index; LVEF, left ventricular ejection fraction; RAS, renin–angiotensin system.

## Data Availability

The data that support the findings of this study are available from the corresponding author upon reasonable request. The data are not publicly available due to internal restrictions on their containing information that could compromise the privacy of research participants.

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
