# Peer review of "Influence of Socioeconomic Status on the Presence of Obstructive Coronary Artery Disease and Cardiovascular Outcomes in Patients Undergoing Invasive Coronary Angiography"

_healthcare, 2024, doi:10.3390/healthcare12020228_

Round 1

Reviewer 1 Report

Comments and Suggestions for Authors

This single-center, retrospective study by Chung et al. examines the relationship between socioeconomic status and incidence of cerebrovascular and cardiovascular events in a high-risk population from South Korea. The main finding is that a low socioeconomic status increases significantly the prognostic value of established risk factors for major adverse cardiovascular events.

Although the topic of this study has been investigated previously, the manuscript’s originality lies in the ethnic background of the target population and in the fact of being a group at high cardiovascular risk.

The manuscript is well organized and easy to read and the figures are acceptable. I have only a few remarks:

Page 3, lines 96-99. If I understand correctly, the assessment of the socioeconomic status of the participants was exclusively based on being beneficiaries of insurance programs, which in turn was based on the individual income. Education was not taken into consideration, yet the number of years spent at school is an important measure of socioeconomic status, and well correlated with cardiovascular risk. The analysis carried out does not allow to ascertain the independent effect of schooling on the outcome. Any comment?

Page 6. According to Table 4, high blood pressure does not appear to be an independent risk factor for cerebrovascular and cardiovascular events. This is strange considering that previous studies have widely shown that hypertension is among the most important cardiovascular risk factors. Couldn’t it be due to the fact that patients were retrospectively defined as hypertensive when they were taking antihypertensive drugs, irrespective of actual pressure values? This could create a bias in the interpretation of the results of Cox regression.

Minor remarks:

In general, the English text is excellent and there are very few misspellings. The most obvious one is at page 8, line 231: “evets” instead of events.

Abstract, line 18. Why “despite?” This preposition does not seem appropriate as it is stated that the prevalence of risk factors was comparable, and the rates of adverse outcomes were also similar.

Page 3, line 134. As far as males are concerned the sum of the numbers in table 1, i.e. 822 + 4918 is 5740, not 5731. Please clarify.

Page 4, Table 1. The obesity percentages within parentheses should be 535/1436 = 37.3% and 3234/8094 = 40.0% unless there are missing values. Please clarify.

Page 9, line 255. Perhaps, “components”.

Comments on the Quality of English Language

Minor editing of English language is required

Author Response

Dear editor,

I, on behalf of all the authors, would like to thank you and the reviewers of Healthcare for taking the time and effort to review our manuscript, entitled “Influence of socioeconomic status on the presence of obstructive coronary artery disease and cardiovascular outcomes in patients undergoing invasive coronary angiography” [Manuscript ID: healthcare-2793264]. Your editorial staffs have provided us with a comprehensive review. Many of the valuable and constructive points that the reviewers pointed out were well taken by all the authors. After going over the reviewers’ comments, my colleagues and I made revisions and indicated the corrections and changes in yellow in the manuscript in the hope of improving our paper.

We hope that our revisions meet the editor’s and reviewers’ expectations. We believe that the comments have significantly improved the quality of our manuscript and hope you will find our revised manuscript acceptable for publication.

We have also attached specific point-by-point responses to the reviewers’ comments as separate files.

Sincerely yours,

Hack-Lyoung Kim, MD, PhD,

Division of Cardiology, Department of Internal Medicine, Boramae Medical Center, Seoul National University College of Medicine, Seoul, 07061, Republic of Korea;

email: [email protected]

Response to the comment from the Reviewer #1:

This single-center, retrospective study by Chung et al. examines the relationship between socioeconomic status and incidence of cerebrovascular and cardiovascular events in a high-risk population from South Korea. The main finding is that a low socioeconomic status increases significantly the prognostic value of established risk factors for major adverse cardiovascular events. Although the topic of this study has been investigated previously, the manuscript’s originality lies in the ethnic background of the target population and in the fact of being a group at high cardiovascular risk. The manuscript is well organized and easy to read and the figures are acceptable. I have only a few remarks:

Response: We appreciate your positive response to our manuscript. We have tried our best to respond to your comments.

Page 3, lines 96-99. If I understand correctly, the assessment of the socioeconomic status of the participants was exclusively based on being beneficiaries of insurance programs, which in turn was based on the individual income. Education was not taken into consideration, yet the number of years spent at school is an important measure of socioeconomic status, and well correlated with cardiovascular risk. The analysis carried out does not allow to ascertain the independent effect of schooling on the outcome. Any comment?

Response: Thank you for your insightful comment. Educational level is indeed an important facet of socioeconomic status (SES). Regrettably, specific information on educational levels is not available in our dataset. In Korea, the categorization of medical insurance type is determined through a complex process that comprehensively reviews not only income level but also various other factors. According to a study that compared and analyzed the characteristics of Medical Aid beneficiaries (MAB) and National Health Insurance beneficiaries (NHIB) in Korea, it was confirmed that MAB generally have lower economic and educational levels compared to those with NHIB.1,2 We have included an additional reference regarding this in the methodology section. Furthermore, we have acknowledged the absence of specific educational level data as a limitation of our research.

Added reference in the method section:

In previous studies, compared to NHIB, MAB had lower income, lower education level, less participation in public pension, and lower private health insurance coverage [14,15].

# Newly added reference: 15. Kim, J-H.; Lee, K-S.; Yoo, K-B.; Park, E-C. The differences in health care utilization between Medical Aid and health insurance: a longitudinal study using propensity score matching. PLoS One 2015;10:e0119939. doi: 10.1371/journal.pone.0119939.

Added description in the discussion section as another study limitation:

Fourthly, our study data did not include specific details for each factor of SES, such as income and educational levels, and lifestyle factors.

Reference:

  1. 1. Kong, N.Y.; Kim, D.H. Factors influencing health care use by health insurance subscribers and medical aid beneficiaries: a study based on data from the Korea welfare panel study database. BMC Public Health 2020, 20, 1133, doi:10.1186/s12889-020-09073-x.
  2. 2. Kim, J-H.; Lee, K-S.; Yoo, K-B.; Park, E-C. The differences in health care utilization between Medical Aid and health insurance: a longitudinal study using propensity score matching. PLoS One 2015;10:e0119939. doi: 10.1371/journal.pone.0119939.

Page 6. According to Table 4, high blood pressure does not appear to be an independent risk factor for cerebrovascular and cardiovascular events. This is strange considering that previous studies have widely shown that hypertension is among the most important cardiovascular risk factors. Couldn’t it be due to the fact that patients were retrospectively defined as hypertensive when they were taking antihypertensive drugs, irrespective of actual pressure values? This could create a bias in the interpretation of the results of Cox regression.

Response: We recognize and appreciate your concerns. As you indicated, the retrospective nature of our study could lead to inaccuracies in defining hypertension. Moreover, we cannot disregard the potential impact of antihypertensive medications. As a result, the well-known adverse effects of hypertension might not have been clearly observable in the multivariable analysis. To address this, we have included further details regarding this matter in the limitations section of our study.

Added sentence in the study limitation section:

Fifthly, our study revealed that hypertension, a well-known CVD risk factor, was not associated with MACCE occurrence in the multivariable analysis. The potential inaccuracy in defining hypertension due to the retrospective design of the study, along with the influence of antihypertensive medications, might have contributed to these unexpected results.

Minor remarks:

In general, the English text is excellent and there are very few misspellings. The most obvious one is at page 8, line 231: “evets” instead of events.

Response: We appreciate your comment, and corrected the typical error.

Abstract, line 18. Why “despite?” This preposition does not seem appropriate as it is stated that the prevalence of risk factors was comparable, and the rates of adverse outcomes were also similar.

Response: We value your feedback. An error was found in the sentence, and it has been amended accordingly.

Despite a higher prevalence of cardiovascular risk factors, the MAB group exhibited similar rates of obstructive CAD compared to the NHIB group.

Page 3, line 134. As far as males are concerned the sum of the numbers in table 1, i.e. 822 + 4918 is 5740, not 5731. Please clarify.

Response: We are very grateful for the thorough review, and the authors have confirmed that there were errors in the Table 1. It seems that the percentage values according to frequency are incorrect because errors were made while calculating them manually. In Table 1, the number of males in the MAB group was 821 (57.2%) and the number of males in the NHIB group was 4,910 (60.7%). The errors were corrected.

Page 4, Table 1. The obesity percentages within parentheses should be 535/1436 = 37.3% and 3234/8094 = 40.0% unless there are missing values. Please clarify.

Response: We are very grateful for the thorough review. It seems that the percentage values according to frequency are incorrect because errors were made while calculating them manually. The errors were corrected.

Page 9, line 255. Perhaps, “components”.

Response: We appreciate your comment, and corrected the typical error.

Reviewer 2 Report

Comments and Suggestions for Authors

This is a good study that can be published given the following amendments and clarifications:

1. Please use an official email address for KHL.

2. Abstract: "Despite a comparable prevalence of cardiovascular risk factors, the MAB group exhibited similar rates of obstructive CAD compared to the NHIB group". Why do you use the word Despite when the two observations are not contradictory? It's only natural that similar risk factors equate to similar CAD rates!

3. While the study shows an association between SES and cardiovascular events, it fails to address what it is about SES that predisposes patients to develop those events. Low SES perhaps entails poor dietary habits and lifestyle choices.

4. Methods: "the following laboratory data". Replace data with parameters.

5. Why did the authors decide to use parametric tests (e.g., t- test)? Did they test the data for normality? It is highly likely that the data are skewed and nonparametric tests (e.g., Mann-Whitney) are thus more appropriate. If this turns out to be indeed the case, then the authors should show medians + IQR instead of mean + SD. 

6. The authors must perform calibration curves and decision curve analysis (DCA) on select parameters that show strong correlation with cardiovascular events.

7. The data shows that DM carries the greatest risk for cardio- and cerebrovascular events (Table 4). This is rather alarming, and the authors should discuss recent advancements regarding novel hyperglycemic markers (PMID: 36816730, 36421613).

8. Minor English editing required. Please see examples on line 85 (serving as a supplementary = serving a supplementary), line 291 (lead worse = lead to worse), and others.

Comments on the Quality of English Language

Minor editing of English language required.

Author Response

Dear editor,

I, on behalf of all the authors, would like to thank you and the reviewers of Healthcare for taking the time and effort to review our manuscript, entitled “Influence of socioeconomic status on the presence of obstructive coronary artery disease and cardiovascular outcomes in patients undergoing invasive coronary angiography” [Manuscript ID: healthcare-2793264]. Your editorial staffs have provided us with a comprehensive review. Many of the valuable and constructive points that the reviewers pointed out were well taken by all the authors. After going over the reviewers’ comments, my colleagues and I made revisions and indicated the corrections and changes in yellow in the manuscript in the hope of improving our paper.

We hope that our revisions meet the editor’s and reviewers’ expectations. We believe that the comments have significantly improved the quality of our manuscript and hope you will find our revised manuscript acceptable for publication.

We have also attached specific point-by-point responses to the reviewers’ comments as separate files.

Sincerely yours,

Hack-Lyoung Kim, MD, PhD,

Division of Cardiology, Department of Internal Medicine, Boramae Medical Center, Seoul National University College of Medicine, Seoul, 07061, Republic of Korea;

email: [email protected]

Response to the comment from the Reviewer #2:

This is a good study that can be published given the following amendments and clarifications:

Response: We appreciate your positive response to our manuscript. We have tried our best to respond to your comments.

  1. Please use an official email address for KHL.

Response: The official email address linked to the university of the corresponding author is "[email protected]". However, for its efficacy and ease in international communication, "[email protected]" is predominantly utilized. Errors occasionally arise when employing the official email for international correspondences. We kindly request your understanding and permission to continue using our current email address.

  1. Abstract: "Despite a comparable prevalence of cardiovascular risk factors, the MAB group exhibited similar rates of obstructive CAD compared to the NHIB group". Why do you use the word Despite when the two observations are not contradictory? It's only natural that similar risk factors equate to similar CAD rates!

Response: We value your feedback. An error was found in the sentence, and it has been amended accordingly.

Despite a higher prevalence of cardiovascular risk factors, the MAB group exhibited similar rates of obstructive CAD compared to the NHIB group.

  1. While the study shows an association between SES and cardiovascular events, it fails to address what it is about SES that predisposes patients to develop those events. Low SES perhaps entails poor dietary habits and lifestyle choices.

Response: We are grateful for your insightful comment. Lifestyle factors are indeed a significant aspect of SES. Unfortunately, our dataset does not include specific information regarding lifestyle factors. We have acknowledged this by adding an additional note to the study limitations section.

Fourthly, our study data did not include specific details for each factor of SES, such as income and educational levels, and lifestyle factors.

  1. Methods: "the following laboratory data". Replace data with parameters.

Response: We changed the word “data” to “parameters”.

  1. Why did the authors decide to use parametric tests (e.g., t- test)? Did they test the data for normality? It is highly likely that the data are skewed and nonparametric tests (e.g., Mann-Whitney) are thus more appropriate. If this turns out to be indeed the case, then the authors should show medians + IQR instead of mean + SD.

Response: We appreciate your feedback. Following your comment, we conducted a comprehensive assessment of all the variables and identified that several of them did not follow a normal distribution. As a result, we opted to analyze these variables using the Wilcoxon rank-sum test, a statistical method, as an alternative to the Student t-test. Additionally, we modified the representation of these variables to "median ± IQR" instead of "mean ± SD." Even with this change, the difference between the two groups shown in the existing data remained the same. The revised portions were marked in red in the method and result sections.

  1. The authors must perform calibration curves and decision curve analysis (DCA) on select parameters that show strong correlation with cardiovascular events.

Response: In response to the reviewer's comment requesting calibration curves and decision curve analysis (DCA) on select parameters with strong correlations to cardiovascular events, we appreciate the suggestion and acknowledge the potential value of these analyses in assessing predictive models. However, after careful consideration, we believe that these particular statistical methods may not be necessary or aligned with the primary objectives of our study. Our study primarily focuses on examining the associations between SES and cardiovascular events, rather than developing predictive models for cardiovascular events. As such, our research goals are more oriented towards understanding the relationships between these variables rather than creating predictive models. We kindly request understanding from the reviewer and hope they will accept our decision to not perform calibration curves and DCA in this particular study. We remain open to any additional suggestions or modifications that would be more relevant to the scope and objectives of our research. Thank you for your understanding and consideration of our response.

  1. The data shows that DM carries the greatest risk for cardio- and cerebrovascular events (Table 4). This is rather alarming, and the authors should discuss recent advancements regarding novel hyperglycemic markers (PMID: 36816730, 36421613).

Response: We value your valuable feedback. Our study confirmed that diabetes is a significant risk factor for MACCE. Alongside diabetes, other recognized risk factors such as advanced age, smoking, reduced left ventricular function, and obstructive CAD were identified in multivariate analysis as associated with the onset of future MACCE. Given the focus of our study on SES, we believe that addressing well-established findings that do not offer new insights falls outside the scope of our research.

  1. Minor English editing required. Please see examples on line 85 (serving as a supplementary = serving a supplementary), line 291 (lead worse = lead to worse), and others.

Response: We value your comment. We have thoroughly reviewed the entire manuscript and corrected any typographical errors.

Round 2

Reviewer 2 Report

Comments and Suggestions for Authors

Accept.